# L-Carnosine Stimulation of Coenzyme Q10 Biosynthesis Promotes Improved Mitochondrial Function and Decreases Hepatic Steatosis in Diabetic Conditions

**DOI:** 10.3390/antiox10050793

**Published:** 2021-05-17

**Authors:** Cheng Schwank-Xu, Elisabete Forsberg, Magnus Bentinger, Allan Zhao, Ishrath Ansurudeen, Gustav Dallner, Sergiu-Bogdan Catrina, Kerstin Brismar, Michael Tekle

**Affiliations:** 1The Rolf Luft Research Center for Diabetes and Endocrinology, Department of Molecular Medicine and Surgery, Karolinska Institutet, SE-17177 Stockholm, Sweden; elisabete.forsberg@su.se (E.F.); magnus.bentinger@ki.se (M.B.); allan.zhao@stud.ki.se (A.Z.); a_ishrath@yahoo.com (I.A.); gustav.dallner@ki.se (G.D.); sergiu-bogdan.catrina@ki.se (S.-B.C.); kerstin.brismar@ki.se (K.B.); michael.tekle@sll.se (M.T.); 2Department of Endocrinology, Diabetes and Metabolism, Karolinska University Hospital, SE-17177 Stockholm, Sweden; 3Centrum for Diabetes, Academical Specialist Centrum, SE-17177 Stockholm, Sweden

**Keywords:** coenzyme Q, carnosine, diabetes, oxidative stress, hepatic steatosis, oxygen consumption rate

## Abstract

Mitochondrial dysfunction in type 2 diabetes leads to oxidative stress, which drives disease progression and diabetes complications. L-carnosine, an endogenous dipeptide, improves metabolic control, wound healing and kidney function in animal models of type 2 diabetes. Coenzyme Q (CoQ), a component of the mitochondrial electron transport chain, possesses similar protective effects on diabetes complications. We aimed to study the effect of carnosine on CoQ, and assess any synergistic effects of carnosine and CoQ on improved mitochondrial function in a mouse model of type 2 diabetes. Carnosine enhanced CoQ gene expression and increased hepatic CoQ biosynthesis in *db/db* mice, a type 2 diabetes model. Co-administration of Carnosine and CoQ improved mitochondrial function, lowered ROS formation and reduced signs of oxidative stress. Our work suggests that carnosine exerts beneficial effects on hepatic CoQ synthesis and when combined with CoQ, improves mitochondrial function and cellular redox balance in the liver of diabetic mice. (4) Conclusions: L-carnosine has beneficial effects on oxidative stress both alone and in combination with CoQ on hepatic mitochondrial function in an obese type 2 diabetes mouse model.

## 1. Introduction

The beneficial effects of L-carnosine (CRN), a dipeptide (β-alanyl-L-histidine), and coenzyme Q (CoQ), a neutral lipid, have been studied in the context of antioxidant functions with increasing interest in recent years. CRN and CoQ are potent endogenous antioxidants with anti-inflammatory properties, and have a potential role in the preventive treatment of diabetes complications [1]. Our laboratory has previously explored the beneficial effects of CRN in the *db/db* mouse model of type 2 diabetes [2].

It has previously been shown that the *db/db* mouse (a type 2 diabetes mouse model) treated with CRN in drinking water for four weeks improved kidney function with significantly reduced proteinuria and improved vascular permeability [3]. CRN also decreased hepatic expression and circulating levels of insulin-like growth factor binding protein-1 (IGFBP-1), lowered blood glucose and increased circulating insulin and insulin-like growth factor 1 (IGF-I) [4]. Additionally, CRN enhanced wound healing with combined topical application and intraperitoneal injections [5]. Previous research has also shown CRN to attenuate both diabetic disease progression and the development of diabetic nephropathy in leptin-deficient ob/ob mice [6]. CoQ is an endogenously synthesized polyisoprenoid that also serves as a powerful antioxidant [7]. The two rate-limiting enzymes, trans-prenyltransferase and 4-OH-benzoate-prenyltransferase, encoded by COQ1 and COQ2 genes, respectively, regulate de novo CoQ biosynthesis [8]. In humans, CoQ has 10 isoprenoid subunits in its side chain (CoQ10), whereas CoQ9 and CoQ6 are the most abundant variants found in rodents and cerevisiae yeasts, commonly used as model organisms in CoQ research [9]. *Db/db* mice treated with CoQ have shown improvements in parameters pertaining to kidney and mitochondrial function [10]. Long-term oral administration of CoQ to *db/db* mice has been observed to prevent peripheral neuropathy by ameliorating diabetes-associated downregulation of Phospholipase C, thereby reducing neuron loss [11]. In diabetic patients, CoQ treatment significantly improved extracellular redox balance, HbA1c, lipid profiles, oxidized LDL, IGF-1 levels, NK-cell activity and immune system responses [12,13]. Additionally, it has been shown that long-term CoQ treatment reduced mortality in the elderly with and without diabetes [14].

CRN and CoQ, while both potent antioxidants by themselves, are differently distributed in different tissues and organs [15,16]. High concentrations of CRN are present in the skeletal muscle, heart and the nervous system, while smaller quantities are locally synthetized in the kidney, liver, ventricle and lungs [17]. CoQ quantities are most abundant in the heart, skeletal muscles, kidney, liver parenchyme and in the pancreas [8]. Furthermore, CRN is a water-soluble molecule, and thus localized in the cytosol, whereas CoQ is lipid soluble and commonly embedded in membrane phospholipid layers [18]. CRN works as a direct oxygen radical scavenger, whereas CoQ neutralizes radicals by donating two electrons sequentially and becomes oxidized in the process [2]. CRN modulates biological functions and processes through its antioxidant, anti-inflammatory and anti-senescence properties. CRN is therefore currently being studied as a potential therapy agent in conditions associated with increased oxidative stress [18,19,20].

T2DM is associated with obesity, dyslipidemia and liver steatosis [21]. Elevated glucose and free fatty acid (FFA) levels drive formation of reactive oxygen species (ROS) in mitochondria [22,23]. T2DM-associated mitochondrial dysfunction accelerates diabetes disease progression and contributes to diabetes complications [24,25]. This dysfunction is characterized by a relative reduction in cellular oxygen consumption during uncoupling experiments, wherein decreased respiration was shown to be associated with reduced cardiac muscle function [26,27]. Due to increased oxidative stress in T2DM, reserve antioxidant capacity may be insufficient [28]. It is known that increased endogenous antioxidant activity and reduced oxidative stress could potentially benefit patients with T2DM and possibly reduce complication rates [29]. The multiple, partly overlapping and partly analogous known effects of CRN and CoQ with a potential role in preventing diabetes complications prompted us to investigate the effects of CRN on CoQ and its possible synergistic effects on cellular respiration and ROS production.

## 2. Materials and Methods

### 2.1. Cell Cultures and Animal Experiments

HepG2 cells (HB-8065, ATCC, Manassas, VA, USA) were cultured in standard composition Dulbecco’s modified eagle medium (DMEM) with added 5.5 mM or 30 mM glucose, 10% fetal bovine serum (FBS), 100 IU/100 μg/mL penicillin-streptomycin and 2 mM L-glutamine (Life Technologies, Carlsbad, CA, USA). L-Carnosine (CRN) (Sigma-Aldrich, St. Louis, MO, USA) treatments, with CRN dissolved in a 99% ethanol (EtOH) vehicle, were applied once every 48 h for a total of 10 days in a final concentration of 20 mM as established by protocol in previous work performed in our lab [4,5]. The *db*/*db* mouse model of type 2 diabetes mellitus (BKS.Cg-Dock7m+/+LeprdbJ) and heterozygote normoglycemic C57BLKS/J littermates (C57B) (Stocks 000642 and 000662, Charles River, Sulzfeld, Germany) were used for animal experiments. A number of 12-week-old male C57B and *db/db* mice with manifest hyperglycemia were studied with and without CRN treatment for 10 days, *n* = 5 for all groups. Animals had ad libitum access to standard formula chow diet and water. For CRN treated groups, 20 mM CRN supplementation was added to the animals’ water supply. For histological analyses, mouse liver samples were snap-frozen in liquid nitrogen (LiN2), kept in 4% paraformaldehyde (PFA) and embedded in histology-grade paraffin. After subsequent de-paraffinization and dehydration, sections were stained with hematoxylin-eosin. Ethical approvals for all animal experiments conducted in this work were obtained from the North Stockholm Animal Ethics Committee.

### 2.2. RNA Purification and Quantitative Real-Time qPCR

Total RNA was extracted from both mouse tissues and cell culture pellets using a miRNAeasy RNA Extraction Kit (Qiagen, Hilden, Germany). The extracted RNA used in the synthesis of cDNA was analyzed using a high-capacity cDNA Reverse Transcription Kit (ThermoFisher Scientific, Waltham, MA, USA). RT-qPCR analysis was then performed using an ABI Prism 7300SDS qPCR system (Applied Biosystems, Foster City, CA, USA). SYBR Green MasterMix qPCR cocktail was used for this procedure (Thermo Fisher Scientific, Waltham, MA, USA). PBGD was used as a housekeeping gene. Primers and primer sequences used are listed in the table below (Table 1).

### 2.3. [3H] Mevalonate Incorporation Experiments

Mevalonate is an intermediate in the synthesis of both cholesterol and CoQ, and mevalonate incorporation analyses are a versatile method to study the biosynthesis of said molecules. In our experimental setup, (R, S)-5-[3H]-mevalonolactone was synthesized using [3H]-sodium borohydride (15 Ci/mmol, American Radiolabeled Chemicals, St Louis, MO, USA) as previously described [30]. The mice were injected with 100 µCi (50 Ci/mmol) mevalonate dissolved in PBS intraperitoneally 1 h prior to sacrifice and analysis. Metabolic labeling of CoQ in monolayer of HepG2 cultures was performed overnight using [3H]-mevalonate (Sigma-Aldrich, St. Louis, MO, USA). 1 mCi [3H]-mevalonate (3.52 Ci/mmol) dissolved in PBS was added to the culture medium. The medium was subsequently removed, and the plates were then washed twice with cold PBS. Cells were recovered by scraping and subsequently centrifuging. The cell pellets were homogenized in PBS and lipids were extracted for HPLC analysis.

### 2.4. Lipid Extraction and HPLC

HPLC-preparatory lipid extraction, isolation, evaporation, dissolving and separation steps were performed using protocols established by our lab as reported previously [31]. The resulting residue was subsequently re-dissolved in chloroform (Sigma-Aldrich, St. Louis, MO, USA) and placed on a silica column (50 mg/1.5 mL: Extract-Clean, Alltech, Deerfield, IL, USA). For subsequent HPLC analysis of CoQ and cholesterol, a reversed-phase HPLC procedure using a SUPELCOSIL LC-18 column (3 μm, 4.0 × 75 mm) with an LC-18 Supelguard column (Supelco, St Louis, MO, USA) was utilized. Lipid content in the resulting eluent was then monitored at wavelengths 210 nm and 275 nm using a standard UV detector. CoQ6 and dolichol-23 were used as internal standards in accordance with previously published work by our lab [32].

### 2.5. β-oxidation Activity In Vitro and In Vivo

In order to assess the effect of CRN treatment on fatty acid metabolism, the rate of fatty acid β-oxidation was measured in HepG2 cells (ATCC) cultured with 5 mM and 30 mM glucose supplemented standard DMEM, with and without CRN treatment. Peroxisomal fatty acyl-CoA oxidase activity was measured as previously described [33]. H_2_O_2_ was quantified fluorometrically through horseradish peroxidase-catalyzed oxidation of 4-hydroxyphenyl-acetic acid into 6,6′- dihydroxy-(1,1′-biphenyl)-3,3′-diacetic acid.

### 2.6. Oxygen Consumption Rate, Oxidative Stress and ROS Formation

HepG2 cells (ATCC) cultured in standard DMEM were treated with 10 mM CRN, 10 µM CoQ, or a CRN/CoQ combination supplemented to the medium for a total of 10 days. Real-time respirometry assays were performed according to established protocols as previously described, using an Agilent Seahorse XF24 cell analyzer (Agilent, Billerica, MA, USA) [7]. Four baseline oxygen consumption rate (OCR) measurements were made, followed by intra-assay well injections of Oligomycin (1 μM), Carbonyl cyanide-4-(trifluoromethoxy) phenylhydrazone (FCCP; 1 μM) and Antimycin A (1 μM). These treatments inhibit ATP synthase, uncouple the electron transport chain and inhibit mitochondrial complex III, respectively. Resulting OCR values were normalized to total protein content. A total of 4 separate experiments were compiled for final results. 4-hydroxynonenal (4-HNE) and carbonylated protein levels in mouse liver lysates were measured using commercially available ELISA kits (OxiSelect STA- 838 and STA-310, respectively, Cell Biolabs, San Diego, CA, USA). Assays were performed according to the manufacturer’s instructions. ROS formation in HepG2 cells was measured using the proprietary superoxide indicator MitoSox Red (Invitrogen, Eugene, OR, USA) at a 2.5 µM working concentration. MitoSox Red Fluorescence was then measured through FACS using a CyAn ADP302 flow cytometer (Beckman-Coulter, Brea, CA, USA) at a wavelength of ex488 nm. FlowJo V10 FACS analysis software (Flowjo, Ashland, OR, USA) was used for subsequent analysis. Real-time superoxide production through electron spin resonance spectrometry (EPR) was measured using a stable radical-forming cyclic hydroxylamine (1-hydroxy-3-methoxycarbonyl -2,2,5,5- tetramethylpyrrolidine, CMH) spin probe and a 3-Carboxy-2,2,5,5- tetramethyl-1-pyrrolidinyloxy CP radical standard curve (Noxygen, Elzach, Germany) according to manufacturer’s specifications and with a testing protocol as previously described [22]. Experiments were performed using a Bruker E-scan (Bruker, Billerica, MA, USA) EPR spectrometer. HepG2 cells (ATCC) were incubated with 200 µM of CMH spin probe at 37 °C for 30 min and subsequently scraped. Supernatant and cells, pelleted and then resuspended while controlling for standardized amounts of shear stress and vibration, were then frozen in liquid nitrogen. Thereafter, measurements were made according to manufacturer’s specifications. FACS and EPR results are a compilation of 4 separate experiments each.

### 2.7. Statistical Analysis

All data in the results are expressed as mean ± SEM. Comparison among groups was performed using one-way ANOVA followed by Tukey’s multiple comparison post-hoc test. Differences between two groups were analyzed using two-sided Student’s *t*-test. Normality of distribution of all data was analyzed using the Kolmogorov–Smirnov test. A *p*-value of *p* < 0.05 was considered significant. All statistical analysis was performed using GraphPad Prism software (version 8, GraphPad Inc., La Jolla, CA, USA).

## 3. Results

### 3.1. Carnosine Modulates Genes Regulating CoQ Expression and Stimulates CoQ Biosynthesis

CRN treatment significantly increased the expression of the PDSS1 (*p* = 0.022) and COQ2 genes (*p* = 0.0046), necessary for CoQ synthesis, in the liver of *db/db* mice but not in the liver of control mice (Figure 1A,B). Carnosine did not significantly alter the relative levels of CoQ9 (Figure 1C), but increased the biosynthesis of CoQ9, as determined by a 65% increase of mevalonate incorporation into CoQ9 in controls and a 30% increase in the *db/db* group (Figure 1D). Carnosine was thus shown to stimulate both expression of CoQ genes and CoQ biosynthesis in the liver of *db/db* mice.

### 3.2. Carnosine Treatment Reduced In Vitro ROS Formation and Oxidative Stress

Elevated ROS formation is known to cause increased oxidative stress-related damage in diabetes mellitus. We therefore measured ROS formation during normal and high glucose conditions in HepG2 cells using MitoSox Red fluorescent superoxide probe FACS and oxygen radical spin trapping EPR spectrometry. High glucose treatment (HG, 30 mM glucose) induced a significantly increased rate of ROS formation (*p* = 0.039), as seen in increased MitoSox Red/PE-Texas Red output. This effect was subsequently abolished after treatment with CRN (Figure 2A,C). Measurement through EPR Spectrometry showed a similar increase in ROS in HG conditions. In EPR measurements, CRN and CoQ by themselves did not reduce levels of HG-induced excess ROS, but CRN and CoQ in combination was seen to revert ROS production to normal levels (Figure 2B,D). To evaluate whether our treatments could affect manifest deleterious effects of increased ROS, 4-HNE and protein carbonyl levels were measured in mouse liver lysates. Here, CRN treatment resulted in significantly reduced protein carbonyl levels in the *db/db* mice (*p* = 0.044) but not in non-diabetic control mice (Figure 2F). The corresponding effect was not seen in 4HNE levels, indicating a difference between the effects on protein carbonylation and lipid oxidation, respectively.

### 3.3. Carnosine Treatment Enhanced CoQ Effect on Mitochondrial Function

As CoQ is known to have beneficial effects on mitochondrial function, we further studied the effect of CRN on OCR in HepG2 cells using cell respirometry (Figure 3A). CRN treatment did not affect basal respiration rate, whereas CoQ by itself and in combination with CRN significantly increased basal OCR (Figure 3B). The same pattern was seen on the amount of ATP produced as assessed after oligomycin injection (Figure 3C). Carnosine did not further enhance the effect of CoQ on OCR or ATP production. CRN treatment, however, increased mitochondrial respiratory reserve capacity and was potentiated by CoQ, which alone had no effect (Figure 3D). CRN and CoQ by itself did not significantly affect proton leak, while a combination of both CRN and CoQ caused a reduction in the proton leak (Figure 3E).

### 3.4. Carnosine Treatment Decreased Cholesterol Synthesis and Reduced Hepatic Steatosis

The improvement effect seen regarding mitochondrial function as seen in HepG2 cell respirometry further prompted us to study the effect of CRN treatment on histological signs of hepatic steatosis in *db/db* mice, since this condition is associated with excessive hepatic cholesterol synthesis and associated negative consequences in T2DM. CRN treatment markedly reduced the size of lipid droplets in the *db/db* mouse liver preparations when comparing untreated (Figure 4G) and treated (Figure 4H) groups, but had no corresponding effect on the WT controls (Figure 4E,F). CRN treatment was also observed to decrease the level of mevalonate incorporation into cholesterol in the liver of *db/db* mice (Figure 4A), which was correspondingly followed by a reduction in total cholesterol levels in liver lysates (Figure 4B).

### 3.5. Carnosine Treatment Did Not Affect β-Oxidation in db/db Mice

We further studied if the observed reduction of liver steatosis in the *db/db* mice through CRN treatment could be explained by increased β-oxidation. Indeed, in HepG2 cells, CRN significantly increased β-oxidation rates in both normal and high glucose conditions (Figure 5A). Conversely, CRN had no effect in vivo, with *db/db* mice exhibiting generally lower β-oxidation rates than controls (Figure 5B).

## 4. Discussion

In this work, we showed that CRN induces in the liver biosynthesis of CoQ in diabetic conditions and that it directly upregulates the PDSS1 and COQ2 genes, which encode rate-limiting enzymes in endogenous de novo *CoQ* biosynthesis [8]. To our knowledge, the CRN-effected upregulation of these genes has not been previously described. CoQ uptake and distribution to different tissues/organs when exogenously supplemented is very low, at around 2–3%, unless endogenous deficiency exists [2,8]. Upregulating CoQ biosynthesis through this novel CRN-mediated mechanism may improve the effective bioavailability of intracellular CoQ, thus harnessing a more beneficial compartmental distribution of CoQ and its constituent oxidized and reduced forms [34]. We also found that CRN treatment combined with CoQ improved mitochondrial function in HepG2 hepatocellular carcinoma cells and reduced protein carbonylation, a consequence of excess ROS reduction which is involved in many pathological mechanisms related to oxidative stress [35,36].

Increasing evidence in recent literature suggests that oxidative stress plays a significant part in the pathogenesis in both type 2 diabetes and its related late-stage complications [37,38,39]. Hyperglycemia causes overproduction of free radicals [40]. CRN treatment has previously been shown to alleviate acetaminophen-induced oxidative stress by upregulating enzymatic and non-enzymatic antioxidants such as catalase, superoxide dismutase and glutathione, as well as TNF-mediated inflammatory increases in superoxide anions [41,42]. Additionally, CRN treatment has also been observed to directly reduce oxidation and glycation products, as well as reduce hepatic steatosis in liver tissue in streptozotocin-induced diabetic rats [43]. Our study did not characterize the specific pathway(s) through which CRN reduces oxidative damage in HepG2 cells and mouse livers, and it is certainly plausible that the effects observed are partly due to known mechanisms involving CRN upregulation of NEF2L2 and inhibition of H_2_O_2_-mediated P38 MAPK-activation [44]. CoQ is known to inhibit lipid peroxidation by preventing the formation of lipid peroxyl radicals [45]. Furthermore, CoQ treatment has been shown to modulate anti-inflammatory processes [13], lower oxidative stress in obese mice [46], and improve mitochondrial function and glycemic control in T2DM [47,48]. Whether our results, showing that there seems to be a synergistic effect between CRN stimulation of CoQ biosynthesis and their respective antioxidative properties, is specifically mediated through upregulated CoQ biosynthesis requires further investigation.

In the present study, CRN enhanced the effect of CoQ on mitochondrial function; thus, the combination of carnosine and CoQ resulted in an increased respiratory reserve capacity and reduced proton leak compared to control or CoQ alone. CRN treatment alone did not show any effect on OCR in HepG2 cells, and previous research has shown that CRN can promote the opposite effect of CoQ upon cellular bioenergetics [49,50]. The synergistic effects seen in the respirometry results also suggest that mitochondria, when the bioenergetic structures are fully saturated with CoQ, may be more responsive to CRN treatment. Previous studies have shown increased oxidative stress and impaired liver mitochondrial function in *db/db* mice [51,52,53]. In our study, carnosine was seen to significantly reduce hepatic fat accumulation in obese *db/db* mice. The mechanism behind this effect could be multifactorial; one could be increased usage of fatty acids in the Krebs cycle or induced peroxisomal β-oxidation activity [54]. In a recent study, carnosine treatment in rats decreased hepatic steatosis and lipid peroxidation but not hypertriglyceridemia, whereas a combination of carnosine and α-tocopherol showed additional effects such as decreased inflammation and insulin resistance [55]. A third explanation is that carnosine reduces lipid formation in liver and adipose tissue as in high fat diet-fed mice [56]. However, CRN is also known to effect bioenergetic substrate-mediated effects independently of oxidative phosphorylation itself, thus illuminating the need for further investigation into underlying mechanistic explanations for our findings [57].

Our study showed that CRN treatment does not change the weight of the *db/db* mice but nonetheless shows beneficial effects on *db/db* mice liver metabolism partly by increasing CoQ biosynthesis, improving mitochondrial function and reducing total cholesterol. These findings suggest a novel link between CRN treatment and an enhanced effect of CoQ, resulting in improved mitochondrial function and reduced ROS production, thus ameliorating oxidative stress and modulating liver steatosis progression. However, the present study does have an acknowledged weakness in the lack of an identified mechanistic explanation behind the observed effects. Beta oxidation was not seen to be affected in the *db/db* mice as a result of treatment despite excessive loss of fat droplets in the liver after carnosine treatment as seen in histological samples. This observation could be due to leptin resistance in the *db/db* mice causing deficient PPAR activity [58]. However, as reported recently, carnosine suppresses both the activity and mRNA expression of fatty acid synthase, HMG-CoA reductase, SREBP-1c and SREBP-2, all coding for enzymes heavily involved in hepatic de novo lipogenesis [56]. The contradictory notion of unaffected beta oxidation and diminished fat droplets in the liver could thus be explained by decreased lipogenesis, as has been previously suggested [59]. Additionally, increased CoQ activity through supplementation or stimulated biosynthesis is associated with reduced cholesterol biosynthesis in the rat liver [46,60,61,62], suggesting that the effect of carnosine could be partly mediated by increased CoQ activity. Further investigation into the mechanistic link between carnosine and its direct effects on lipogenesis are planned in order to elucidate the potential use of CRN as a potential therapeutic agent for conditions associated with absolute or relative lack of CoQ.

## 5. Conclusions

Our study showed that L-carnosine enhanced the biosynthesis and antioxidative effects of CoQ, resulting in improved hepatic mitochondrial function and reduced ROS production in vitro and in vivo. The role of impaired mitochondrial function and oxidative stress in fatty liver and the role of antioxidants therein warrants further study. Carnosine, and especially a combination of carnosine and CoQ, is thus identified as a potential candidate for protection against NAFLD through improving mitochondrial function, and reducing oxidative stress, hepatic insulin resistance and lipogenesis.

## Figures and Tables

**Figure 1 antioxidants-10-00793-f001:**
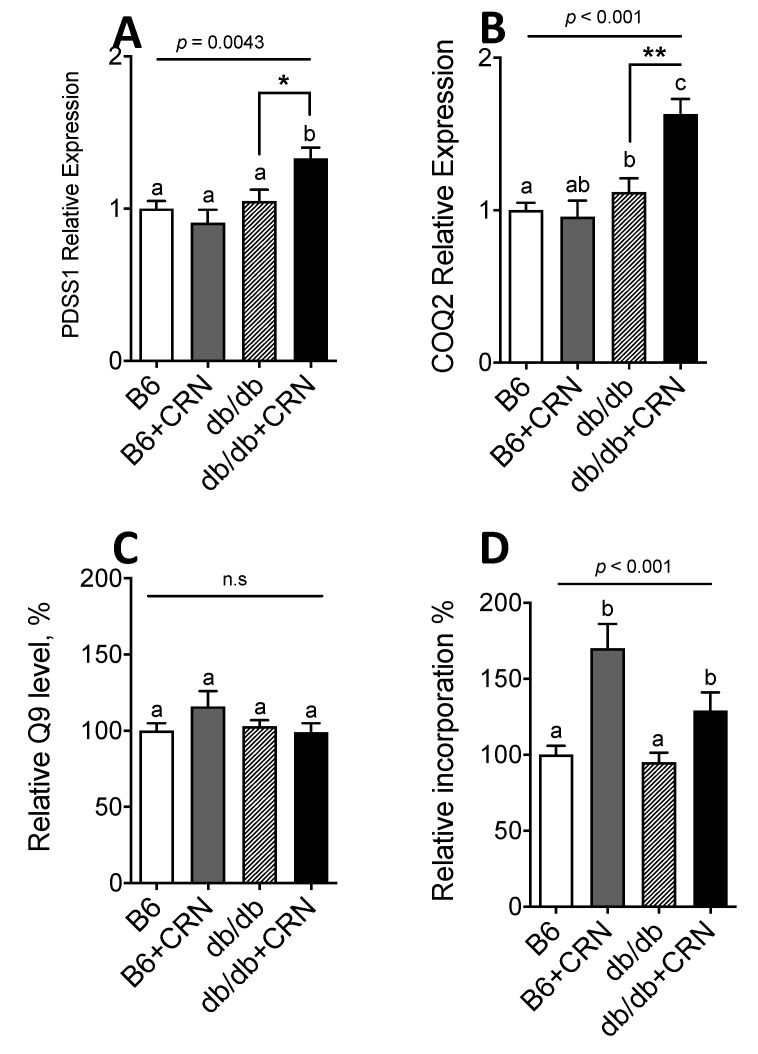
PDSS1 (**A**) and COQ2 (**B**) gene expression measured in 12-week-old C57B and *db/db* mice, with and without treatment of CRN at a 20 mM final concentration mixed with freely available drinking water during a treatment period of 4 weeks. Total CoQ9 levels in liver tissue from the same *db/db* and control mice treated with sham or CRN (**C**). Quantification of [3H]-mevalonate incorporation into CoQ9 in liver tissue from control (C57B) and *db/db* mice treated with sham or 20 mM CRN in drinking water for 4 weeks (**D**). PBGD was used as a housekeeping gene for all gene expression analyses. Animal experiments were *n* = 5 for each condition unless stated otherwise. Values are represented as mean ± SEM in each group. Statistical analysis was performed using one-way ANOVA with multiple comparison differences denoted. (a,b) Different letters indicate significant differences in mean values from repeated-measures ANOVA (*p* < 0.05). A *p*-value of <0.05 was considered significant. * and ** denotes a significance of *p* < 0.05, *p* < 0.01, respectively in all cases.

**Figure 2 antioxidants-10-00793-f002:**
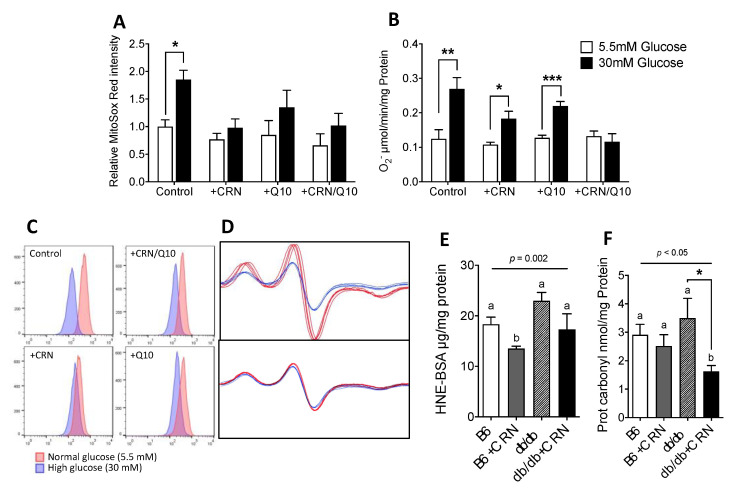
FACS results, expressed as median PE-Texas Red emission at ex488nm in HepG2 cells labeled with MitoSox Red. Cells were either treated with sham, 10 mM CRN, 10 µM CoQ or a combination of CRN and CoQ at 10 mM and 10 µM, respectively. Results are presented as values relative to control (**A**). Representative FACS histograms of MitoSox Red labeled HepG2 cells as presented (**C**). Superoxide production in HepG2 cells measured using CMH spin trapping in EPR spectroscopy (**B**). FACS and EPR experiments are compiled results from 4 separate experiments. Results from EPR were normalized to total protein content. EPR spectra for cells treated with sham and a 10 mM CRN/10 µM CoQ combination shown (**D**). 4-Hydroxynonenal content in mouse liver lysates measured by 4-HNE adduct competitive ELISA (**E**) for treated and untreated control (C57B) and *db/db* mice. Protein Carbonyl ELISA results from the same animals (**F**). ELISA results were normalized by total protein content. *n* = 5 for all groups. All values are represented as mean ± SEM. Statistical analysis was performed using Student’s *t*-test (**A**,**B**) and one-way ANOVA with multiple comparison differences denoted (**E**,**F**). (a,b) Different letters indicate significant differences in mean values from repeated-measures ANOVA (*p* < 0.05). A *p*-value of <0.05 was considered significant. *, ** and *** denotes a significance of *p* < 0.05, *p* < 0.01 and *p* < 0.001, respectively in all cases.

**Figure 3 antioxidants-10-00793-f003:**
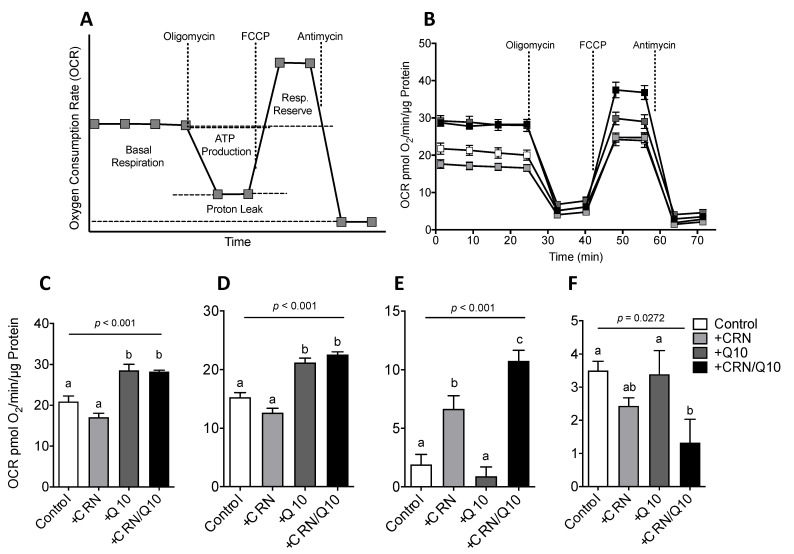
Schematic diagram of a standard OCR readout plot with output parameters pertaining to oxphos metabolism labeled (**A**); Sample HepG2 XF24 respirometry assay OCR readout from a single experiment with intra-repeat variation (**B**); Basal respiration OCR levels, inferred through the average of four baseline measurements minus non-mitochondrial respiration levels (**C**); ATP production, calculated as the difference between baseline respiration and OCR after oligomycin injection (**D**); Respiratory reserve, calculated as the difference between maximal OCR after FCCP injection and basal respiration (**E**); Proton leak, calculated as the difference between OCR after oligomycin injection and residual non-mitochondrial respiration measured as OCR after Antimycin A injection (**F**). Results are normalized to total protein content per well and expressed as pmol O2/min/µm protein. Results are compiled from 4 separate experiments. All values are represented as mean ± SEM. Statistical analysis was performed using one-way ANOVA with multiple comparison differences denoted. (a,b) Different letters indicate significant differences in mean values from repeated-measures ANOVA (*p* < 0.05). A *p*-value of <0.05 was considered significant.

**Figure 4 antioxidants-10-00793-f004:**
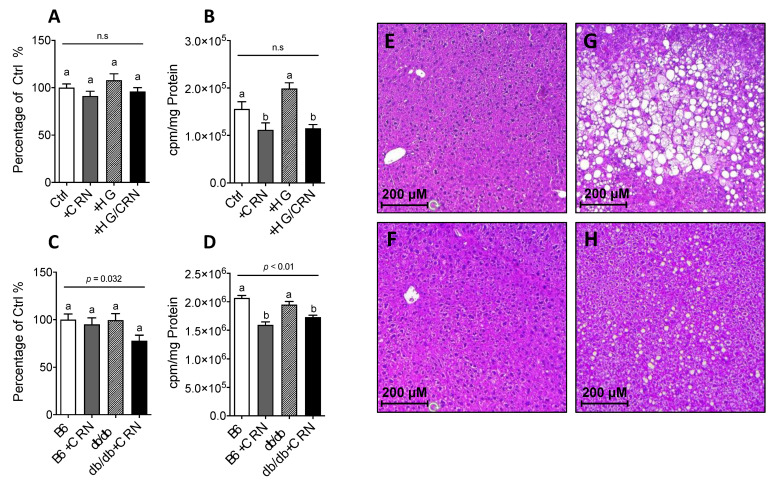
Quantification of total amounts of cholesterol in HepG2 cells with and without 10 mM CRN treatment in 5.5 mM or 30 mM glucose (**A**). Cholesterol biosynthesis in HepG2 cells treated with and without 20 mM CRN in 5.5 mM or 30 mM glucose (**B**). Total cholesterol levels in the livers of control and *db/db* mice with and without 10 mM CRN treatment (**C**). Radioactive labelled mevalonate incorporation into cholesterol in liver tissue of CRN treated and untreated control (C57B) and *db/db* mice (**D**). Animal experiments were *n* = 5 for each condition unless stated otherwise. Values are represented as mean ± SEM. in each group. Statistical analysis was performed using one-way ANOVA with multiple comparison differences denoted. A *p*-value of <0.05 was considered significant. Liver cryosection histology showing representative microscopy photographs at 10× depicting steatotic fat droplets with hematoxylin and eosin staining (**E**–**H**). Sections from each treatment group are shown; upper left panel, untreated control (**E**), lower left panel control + CRN (**F**), upper right panel, untreated *db/db* (**G**), and lower right panel *db/db* + CRN (**H**). CRN treatment reduced droplet size and quantity in *db/db* mice, thereby alleviating pathological signs of hepatic steatosis. Animal experiments were *n* = 5 for each condition unless stated otherwise. The results shown are the mean of three different experiments performed with five animals in each group. Values are represented as mean ± SEM in each group. (a,b) Different letters indicate significant differences in mean values from repeated-measures ANOVA (*p* < 0.05).

**Figure 5 antioxidants-10-00793-f005:**
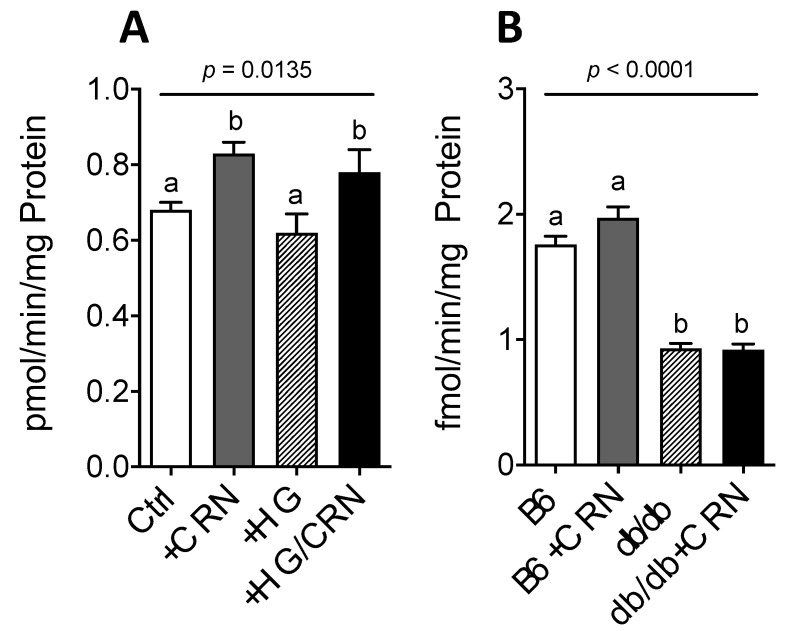
β-oxidation rate in HepG2 cells grown in normal (5.5 mM) and high (30 mM) glucose with and without treatment with 10 mM CRN, 10 µM CoQ or a combination of the two (**A**). β-oxidation rate in control (C57B) and *db/db* mice liver treated with CRN, CoQ or CRN+CoQ compared to non-treated controls (**B**). The results represent mean values of five different experiments performed in triplicate. Values are represented as mean ± SEM in each group. Statistical analysis was performed using one-way ANOVA with multiple comparison differences denoted. (a,b) Different letters indicate significant differences in mean values from repeated-measures ANOVA (*p* < 0.05). A *p*-value of <0.05 was considered significant.

**Table 1 antioxidants-10-00793-t001:** List of primer sequences used for RT-qPCR.

Gene	Forward Primer (5′→3′)	Reverse Primer (5′→3′)
PDSS1 (Human)	5′-TCA TAG GCG GAA GGG ACT TGA-3′	5′-GGT TGT GTG ATG AAA CCG TGA T-3′
PDSS1 (Mouse)	5′-CGG TTC AGT TTG CCA GGA GAT-3′	5′-GCG TCC CTT TCT GTA GAT GGT-3′
COQ2 (Human)	5′CGG TTG GCA AAG CCC ATT G-3′	5′-GGA CGA TTG GCT GTT CTT GTA-3′
COQ2 (Mouse)	5′-ACA AGC CCA TAG GAA CCT GG-3′	5′-CTC CAC GCA TCA GAA TAG CTC-3′
PBGD (Human)	5′-AGG ATG GGC AAC TGT ACC-3′	5′-GTT TTG GCT CCT TTG CTC AG-3′
PBGD (Mouse)	5′-ACT CTG CTT CGC TGC ATT G-3′	5′-AGT TGC CCA TCT TTC ATC ACT G-3′

## Data Availability

The data presented in this study are available on request from the corresponding author.

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
