# Peer review of "L-Carnosine Stimulation of Coenzyme Q10 Biosynthesis Promotes Improved Mitochondrial Function and Decreases Hepatic Steatosis in Diabetic Conditions"

_antioxidants, 2021, doi:10.3390/antiox10050793_

Round 1
Reviewer 1 Report
Interesting study, well planned and designed.
Author Response
We thank Reviewer 1 for the feedback and approval. Language edits have been made according to suggestions given.
Reviewer 2 Report
In this study, Schwank-Xu et al. investigated the effect of L-carnosine treatment, combined with coenzyme Q10, on mitochondrial function and hepatic lipid metabolism. HepG2 cells and db/db mice model of diabetes. It seems that experiments were conducted appropriately, but the following concerns should be addressed before considering publication.
Major comments
1. Statistical analysis and results are not appropriately described. According to the method 2.7., the authors used ANOVA followed by Tukey's test. However, Fig. 1 seems to compare Ctrl vs. CRN and HG vs. HG/CRN. The t-test was used here? or not all Tukey's test results are shown? Please clarify. Fig. 2A and 2B were probably analyzed by t-test, but it is not described. Asterisks in Fig. 3 are not clearly showing the comparison. For example, Fig. 3D * shows the P value between CRN and CRN/Q10, but *** is the P value between what and what? I suggest to use alphabets to show statistical significance for multiple comparisons. There must be many examples but see Fig. 2 of Comparative Biochemistry and Physiology, Part A 161 (2012) 145–152 as an example.
2. Fig. 2A: I did not understand the meaning of “This effect was reversed after treatment with CRN” in L165. Did the authors mean that the significant increase of MitoSox Red intensity became non-significant? If so, “reversed” should be something like "abolished the significant increase".
3. More importantly, I do not agree with the statement “CoQ had no effect on ROS formation” in L166. In Fig. 2A the effect of Q10 is quite similar to that of CRN.
4. Introduction and Discussion do not provide sufficient information. Increase in PDSS1 and COQ2 genes by carnosine is firstly reported in this paper? There are some compherensive gene expression analyses about the effect of carnosine. The effects of this study should be planned or explained taking more studies into account. At least references 3-7 are all from the authors' group (overlapping coauthors).
5. Title: “ameliorates hepatic steatosis” in the title is a little too much based on the data in Fig. 4. More specific word usage is encouraged.
Minor comments
Title: CoQ is used in the text, whereas Q10 is used in the title and figures. Better to be consistent.
L52: “tissues and cellular compartments”: L52-55 are all about tissue distribution. Description about cellular compartments starts at L55, which is far from this topic sentence.
L248: exists
Author Response
We thank the reviewer for very insightful and good comments.
Answer to comment 1: As stated in the methods, only one-way ANOVA with Tukey’s post-hoc test was used. For Fig.1, not all columns were compared with Tukey’s, but only between the Ctrl/CRN group as well as HG/HG/CRN group. It is possible that significance is shown when comparing more groups, but it is not included in this specific analysis as we only look in the difference within the groups. Furthermore, a t-test would not be appropriate here, since all 4 groups are displayed in the same graph.
For Fig.3, asterisks above the columns show significance to baseline. We apologize for the ambiguity and have clarified this in the text for all figures.
Answer to comment 2: We thank the reviewer for this comment and have changed the result text pertaining to fig 2A accordingly.
Answer comment 3: The reviewer is correct in that no conclusions can be drawn regarding whether the reduction in MitoSox red activity is due to CRN and Q10 in combination or CRN and/or Q10 alone. The contrary claim has been removed in the text.
Answer comment 4: The discussion section has been revised and fleshed out in accordance with the critique provided. To our knowledge, this is indeed the first time a direct upregulation of PDSS1 and COQ2 due to carnosine treatment has been reported.
Answer to comment 5: We thank the reviewer for this comment and have changed the title to “Decreases” instead of “Ameliorates”.
Answer to minor comments: The usage of CoQ instead of Q10 in the text is a conscious choice made by the authors in order to have a catch-all term incorporating CoQ10 (humans), CoQ9 (rodents) and CoQ6 (yeast) in instances where the general meaning of “Coenzyme Q10 in the applicable biological context” is intended. Whereas usage of Q10 and Q9 in figure texts refer to specific molecules. The inconsistency seen in figure legends for fig.2 has been amended to reflect this intent.
Reviewer 3 Report
This study explores the effects of L-Carnosine stimulation of coenzyme Q10 biosynthesis and improved mitochondrial function and ameliorates hepatic steatosis in diabetic conditions, but these research analysis items are too simple to understand the possible molecular mechanism of the improved mitochondrial functio. Based on the above reasons, it is recommended that the author further investigates the protein or RNA expressions in the liver or heart for understanding the possible mechanism. I still have some major comments below: 1.the expression and description on method 2.5 should be re-written. 2.What is RQ? in the method 2.5? 3.XF24?(line 118) 4.the author should provide abbreviations list 5.CRN is insoluble in water and had low bioavailability, and how author dissolve CRN into medium?Author Response
We thank the reviewer for his good and insightful comments.
Answer to question 1: The title of Method section 2.5 is re-written and corrected (113).
Answer to question 2: RQ is an abbreviation for Respiratory Quotient. We apologize for not making it clear. Due to redundancy the term has been redacted (L113).
Answer to question 3: XF24 is a specific Seahorse Analyzer made from Agilent. This has been clarified in the text (L123-124).
Answer to question 4: We apologize for the ambiguity of unexplained abbreviations and have made sure to have clear descriptions of what every abbreviation means in the text.
Answer to question 5: CRN as used was dissolved in an ethanol-based vehicle and subsequently added to cell media.
Further investigations are indeed planned with RNA sequencing being one option being assessed. As for now however the time frame allotted for the present special issue of Antioxidants, and conditions provided by external circumstances only allow for us presenting our research on a more descriptive level at present.
Round 2
Reviewer 2 Report
The revised manuscript shows some improvements. Unfortunately, however, the authors did not address my concern about statistics. Indeed, I cannot understand the reason why the authors ignore a comments from a reviewer in this way without providing any rationale. It is not appropriate to show the comparison only you are interested in (in all figures). Consider my suggestion to use alphabets. If the authors want to compare only Ctrl - CRN and HG - HC/CRN, there is no reason to use ANOVA and Tukey test. The comment from the authors “Furthermore, a t-test would not be appropriate here, since all 4 groups are displayed in the same graph” is not correct either. t-test is not appropriate for multiple comparison, no matter how the graph looks like.
Author Response
We thank the reviewer for pointing out areas of improvements in our manuscript, and sincerely apologize for not having addressed all comments sufficiently in the previous revision. The stylistic choice of highlighting significant findings in multiple comparisons while leaving out other results were perhaps misguided, and we have now revised the graphs to show an alphabetical significance notation without specific significance tiers/multiple asterisks. Relevant significances denoted with asterisks now have their p-values reported in the main text.
We have thus included all the multiple comparisons that are significant in all the figures. We have also elected to analyze and present Figure 2A and 2B with the t-test for treatment effect, which is to be seen as the relevant finding for those respective experiments (FACS and EPR). Figure legends have been revised to clarify which statistical test that was used in each individual case.
Once again, we thank the reviewer for the comments, and agree that the suggested changes make for a easier understanding of the paper’s core findings.
Reviewer 3 Report
The author had corrected the manuscript according to my comments.
Author Response
We thank the reviewer for the approval given, and have made final adjustments to the main text, figures and legends.